# Digestate Improves Stinging Nettle (*Urtica dioica*) Growth and Fiber Production at a Chlor-Alkali Site

**DOI:** 10.3390/plants13172425

**Published:** 2024-08-30

**Authors:** Chloé Viotti, Coralie Bertheau, Françoise Martz, Loïc Yung, Vincent Placet, Andrea Ferrarini, Flavio Fornassier, Damien Blaudez, Markus Puschenreiter, Michel Chalot

**Affiliations:** 1Université de Franche-Comté, CNRS, Chrono-Environnement, F-25200 Montbéliard, France; chloe.viotti@univ-fcomte.fr (C.V.); coralie.bertheau-rossel@univ-fcomte.fr (C.B.); 2Production System Unit, Natural Resources Institute Finland (Luke), Ounasjoentie 6, 96200 Rovaniemi, Finland; francoise.martz@luke.fi; 3Université de Lorraine, CNRS, LIEC, F-54000 Nancy, France; loic.yung@uha.fr (L.Y.); damien.blaudez@univ-lorraine.fr (D.B.); 4Université de Franche-Comté, FEMTO-ST Institute, Department of Applied Mechanics, F-25000 Besançon, France; vincent.placet@univ-fcomte.fr; 5Department of Sustainable Crop Production, Università Cattolica del Sacro Cuore, Via Emilia Parmense 84, 29122 Piacenza, Italy; andrea.ferrarini@unicatt.it; 6CREA—Centro Viticoltura ed Enologia, Via Trieste 23, 34170 Gorizia, Italy; flavio.fornasier@crea.gov.it; 7Institute of Soil Research, University of Natural Resources and Life Sciences, 1180 Vienna, Austria; markus.puschenreiter@boku.ac.at; 8Université de Lorraine, Faculté des Sciences et Technologies, F-54000 Nancy, France; 9UMR 6249 Laboratoire Chrono-Environnement, Pôle Universitaire du Pays de Montbéliard, 4 Place Tharradin, F-25200 Montbéliard, France

**Keywords:** compost, digestate, enzymatic activities, fiber, marginal lands, N fertilization

## Abstract

Marginal lands have been proposed to produce non-food crop biomass for energy or green materials. For this purpose, the selection, implementation, and growth optimization of plant species on such lands are key elements to investigate to achieve relevant plant yields. Stinging nettle (*Urtica dioica*) is a herbaceous perennial that grows spontaneously on contaminated lands and was described as suitable to produce fibers for material applications. Two mercury-contaminated soils from industrial wastelands with different properties (grassland soil and sediment landfill) were used in this study to assess the potential growth of stinging nettle in a greenhouse mesocosm experiment. Two organic amendments were studied for their impact on nettle growth. The solid digestate from organic food wastes significantly doubled plant biomass whereas the compost from green wastes had a lower impact. The highest doses of organic amendments significantly increased the number of fibers, which doubled following digestate application, while reducing leaf Hg concentration. Both amendments significantly improved soil respiration and enzymatic activities linked to the microbial biomass in the soil from the sediment landfill by the end of the experiment. In the context of a phytomanagement scenario, solid digestate would be a preferred amendment resource to improve nettle production on industrial wastelands.

## 1. Introduction

Due to global population growth, agricultural food production has become more important over the years, putting great pressure on the environment [1]. Other needs linked to the bio-based economy (e.g., bioenergy, biomass, fiber) have emerged and have generated new competition for land use between food and non-food crops since the early 2000s [2,3]. Due to low fertility and high environmental stress, marginal lands are unsuitable for food cultivation but have been proposed as relevant land to produce biomass for energy or green material purposes [4,5]. In Europe, marginal lands cover an estimated area of 39 to 111 million hectares [6,7], and their use for cultivation can meet the needs of a circular economy action plan and bioeconomy strategy from the European Green Deal [8,9].

Among marginal lands, trace element (TE)-contaminated soils have been described as a relevant alternative to produce biomass for material applications [10,11]. In addition to the potential economic benefit of biomass production on these lands [12,13], vegetation cover also provides environmental benefits. For example, it can help to prevent the dispersion of contaminated soil particles, reduce the mobility of metals in the rhizosphere [14], and improve soil properties (e.g., organic matter and nutrient content, biological activity) [15]. Although some TE-tolerant plant species growing or implemented in contaminated or polluted soils are known to accumulate TE [16,17], others such as industrial hemp (*Cannabis sativa* L.) or ramie (*Boehmeria Nivea*) exhibit a low bio-concentration factor in the aerial parts. Combined with a high yield crop, the biomass produced can be used for industrial fiber applications [18,19]. However, biomass production should be sufficient and slightly impacted by soil contamination, with yields almost similar to those on uncontaminated soils [20]. Thus, plant tolerance to abiotic stress (e.g., nutrient deficiency) is the most important consideration for establishing a biomass crop on these lands [21,22]. Under such conditions, modifications of the biosynthesis of compounds through specialized metabolism can be observed in plant parts [23,24]. Among these metabolites, phenolic compounds are often produced as a result of the stress response [25]. These metabolites are abundant in plants and involved in different physiological processes such as growth, stress defense, and antioxidant protection [26,27]. Thus, phenolics are considered to be involved in the adaptation of the plant to its environment [28].

Stinging nettle (*Urtica dioica*) is a herbaceous perennial plant that grows spontaneously on contaminated lands, notably in mercury (Hg)-contaminated soils, and that produces fibers that can be used for material applications [29]. Hg contamination has increased over the last few decades and is of concern because of its toxicity to humans and ecosystems and its potential accumulation by plants [30]. Nettle grown under pot-based laboratory-scale experiments exhibited excess levels of metals, above the toxic levels reported by Kabata-Pendias [31], while plant material collected from in situ contaminated sites had TE levels below toxic levels [32]. Although *U. dioica* is native and able to grow in TE-contaminated soils, its cultivation in the context of biomass and fiber production has not yet been investigated. *Urtica dioica* is usually described as a nitrophilous species [33] for which cultivation requires high doses of nitrogen (N) fertilization [32,34]. As a perennial plant, annual re-establishment is not necessary. A duration of four to five years has been described for a stinging nettle crop [35], to a maximum of 10 years with high weed control [32,36]. Although *U. dioica* competes poorly with weeds [37], Müllerová et al. [38] improved stinging nettle expansion and regeneration in cut grasslands using N fertilization (along with phosphorus (P) and potassium (K)). The use of amendments constitutes a factor that assists plant adaptation by improving nutrient availability, soil physicochemical properties, and microbial activity [39,40,41].

Due to the environmental impacts of inorganic fertilizers [42,43], the use of organic amendments is now encouraged as an alternative to improve both soil fertility and carbon (C) storage [44]. Organic amendments can be produced from agricultural residues or food wastes, and must be rigorously selected as they can have different effects [45,46]. Among these amendments, compost, which results from the stabilization and sanitation of organic waste by aerobic decomposition, is a well-known product [47]. Digestate is another organic amendment, either as solid or liquid material, which is produced under a controlled anaerobic fermentation process from biodegradable materials [48]. The production of such fertilizers from bio-wastes can therefore be part of the bio-based circular economy concept by reincorporating waste materials into the production cycle [49,50], as well as potential substitutes for mineral N fertilizers [51,52].

N cycling in the soil is crucial for plant development and crop yield [53], and N is released from compost and solid digestate slowly over time through mineralization [54,55]. N availability is primarily driven by microbial reactions [56]. Soil microorganisms produce extracellular hydrolytic and oxidative enzymes involved in C, N, and P cycling and therefore participate in the breakdown of organic matter [57,58]. These enzymes can be stimulated or inhibited by N application [59,60] impacting soil nutrient availability [61]. Moreover, compost and digestate, through adsorption mechanisms and enhancement in microbial activity, may also be involved in soil contaminant immobilization and thus plant uptake limitation [62,63]. Their use therefore appears to be relevant in the context of biomass production from contaminated lands.

The present study aimed to investigate the fitness of *U. dioica* growth on Hg-contaminated soils and to determine the most effective organic amendment (compost: C or digestate: D) and dose (+; ++; +++) for biomass production improvement. We conducted the experiment on two industrial soils with different properties, but both were contaminated with Hg. Stinging nettle plants were grown in pots and amended with compost or solid digestate at different doses. Plant parameters (i.e., biomass, height, stem diameter, chlorophyll content, photosystem II (PSII) activity, phenolic compounds, N and Hg concentrations, fiber characteristics) were measured to evaluate the growth of *U. dioica* on these contaminated soils. Soil parameters (i.e., soil respiration, double-stranded DNA (dsDNA) concentration and enzymatic activities) were assessed to identify the effect of organic amendments on soil activities and microbial biomass that can impact plant growth. We therefore hypothesized that the use of organic amendments would (i) improve nettle fitness in Hg-contaminated soils and (ii) sustain biomass production by improving soil microbial biomass and activities.

## 2. Results

### 2.1. Main Impacting Factors and Effect of the Soil on Nettle Growth

All replicates survived during the experiment, with PSII activity that ranged from 0.75 to 0.78 in both soils (Appendix A). The application of amendments significantly decreased the chlorophyll content, except for D++, which resulted in the opposite effect (Appendix A). Leaves (<153 μg·kg^−1^ DW) and stems (≤72 μg·kg^−1^ DW) exhibited low Hg concentrations regardless of the soil considered (Table 1). However, D++ significantly divided the Hg concentration in the leaves of nettles from the Tavazzano soil by 2.3 and the Hg concentration in the leaves and stems of nettles from the St-Symphorien-sur-Saône soil by 1.5.

When all the measured variables were considered, the applied dose of amendment was the main impacting factor (R^2^ = 0.53, *p* = 0.001), followed by the treatment (i.e., nature of amendment × dose × soil; R^2^ = 0.10, *p* < 0.05). The two first principal components explained 59.6% and 51.2% of the total variance in the whole dataset for St-Symphorien-sur-Saône and Tavazzano soils, respectively, with digestate samples that seemed to separate from the control rather than the compost samples (Figure 1a,c). Plant parameters (i.e., aboveground biomass, plant height and leaf area) contributed the most to the variability of the data, along with enzymatic activities involved in N and P cycles (i.e., chit, leu, acP, bisP) for both soils (Figure 1b,d). Except for xilo and uroni, the enzymatic activities involved in the C cycle contributed the least, along with soil respiration and dsDNA for the Tavazzano soil. Finally, the soil had a low significant impact on the variability of the dataset (R^2^ = 0.05, *p* < 0.01). However, control plants grown on the Tavazzano soil produced significantly twice more aboveground biomass than nettles grown on the St-Symphorien-sur-Saône soil (Figure 2a,d). Nettles resulted in significantly higher plants (Figure 2c,f) with a higher stem diameter (Figure 3a,b).

### 2.2. Impact of the Nature and Dose of Amendments on Plant Growth

The D+, D++, and C+++ treatments significantly increased the production of belowground biomass of nettles in the St-Symphorien-sur-Saône soil (Figure 2a). The nature of the amendment and the dose applied mainly affected the final aerial biomass (F = 147.2; *p* < 0.001 and F = 86.3; *p* < 0.001, respectively) rather than the soil (F = 24.7; *p* < 0.001). At harvest, only the plants that grew under the highest dose of digestate reached the flowering stage. In the St-Symphorien-sur-Saône soil, C+++ and D+ significantly doubled the produced aboveground biomass, while D++ more than quintupled it (Figure 2a). During the first four days of cultivation, C+++ and D++ significantly reduced the relative growth compared to control plants, while the positive impact of these treatments started to be significant after 49 days of cultivation in this soil (Figure 2b). From 63 days, D++ significantly improved the relative growth compared to all the other treatments. At harvest, the plant height followed the same increase as above- and belowground biomass, with nettles significantly taller with digestate and C+++ compared to control plants (Figure 2c).

All amendments and doses applied to the St-Symphorien-sur-Saône soil significantly improved the stem diameter, and C+++ and D++ resulted in significantly more and longer internodes (Figure 3a). Concomitantly, the number of fibers increased significantly by 1.3 and 2.4 times under the C+++ and D++ treatments, respectively (Figure 4). Conversely, the fertilization did not significantly impact the fiber diameters, which ranged on average from 37 to 49 µm, or the wall thicknesses (Appendix A). Among the plant parameters measured, plant height largely explained the aboveground biomass (F = 835.1; *p* < 0.001), followed by the leaf area (F = 26.7; *p* < 0.001), the number of leaves (F = 24.2; *p* < 0.001), and stem diameter (F = 9.6; *p* < 0.01).

In the Tavazzano soil, D++ significantly almost doubled the belowground biomass, while the compost application significantly decreased it (Figure 2d). However, only digestate application significantly improved the aerial biomass of nettle (Figure 2d). When applied to the Tavazzano soil, D++ significantly slowed the relative growth during the first eleven days of cultivation before significantly increasing it after 63 days compared to all the other treatments (Figure 2e). At harvest, only D++ significantly doubled the plant height without the impact of the other treatments (Figure 2f). In this soil, only the digestate application had an impact on the stem diameter, and only D++ significantly improved the number and length of internodes (Figure 3b). Among the measured plant parameters, plant height mainly explained the aboveground biomass (F = 782.1; *p* < 0.001), followed by the number of leaves (F = 65.9; *p* < 0.01), which was the second explanatory factor of aerial biomass before leaf area and stem diameter (F = 23.3; *p* < 0.01 and F = 24.8; *p* < 0.01, respectively).

### 2.3. Responses of Soil Activities to Compost and Digestate Application

In the two control rhizospheric soils, alkP, butyr, nona, bisP, leu, and acP enzymatic activities had the highest values (Figure 5a,c; Appendix A). Only the compost application slightly but significantly increased the soil pH of the Tavazzano soil at the end of the experiment compared to the control soil (pH = 7.5 and 7.3 in the soil amended with compost; and pH = 6.9 in the control soil), while other treatments did not impact the pH of the soil (Appendix A). However, amendment application significantly impacted the enzymatic activities, with the highest impact of the dose compared to the low impact of the treatment (nature of amendment × dose × soil) and the soil (R^2^ = 0.39, *p* = 0.001; R^2^ = 0.07, *p* < 0.01; R^2^ = 0.04, *p* < 0.05, respectively). Despite not being the most impacting factor, the Tavazzano control soil exhibited significantly higher enzymatic activities than the St-Symphorien-sur-Saône soil. In both soils, all the amendments, except D+ in the Tavazzano soil, significantly increased butyr (four- and five-fold with C+++ and D++ in the St-Symphorien-sur-Saône soil) and nona (seven-fold increase with the two higher doses of amendments in the St-Symphorien-sur-Saône soil) esterase activities (Figure 5a,c). All the amendments significantly increased the activities of xilo and uroni involved in the C cycle in the St-Symphorien-sur-Saône soil, but the activities remained low (<1 nmol 4-MUF·g^−1^ soil per hour) (Figure 5a). The digestate significantly increased the enzymatic activities involved in the N cycle, which were, respectively, twice (leu in both concentrations and chit with D+) and three times (chit with D++) higher than in the control (Figure 5a). C+++ and D++ resulted in significantly higher piroP and alkP enzyme activities, and the same was observed for acP, bisP, and aryS activities also with C++. However, amendment application had a lower impact on the enzymatic activities in the Tavazzano soil. C+++ significantly increased the activities of alfaG, xilo, alkP, and bisP (Figure 5b, Appendix A). In contrast, the activities of leu, acP, bisP with D+, aryS with C++ and D++, and uroni with digestate and C+ were slightly but significantly decreased compared to the activities in the control soil (Figure 5b).

At the end of the cultivation period, all the amendments except C++ significantly improved the respiration of the St-Symphorien-sur-Saône soil, which doubled with digestate application (Figure 6a). Both amendments significantly increased dsDNA (Appendix A), which was correlated with soil respiration and enzymatic activities involved in C and N cycles (i.e., xilo, uroni, chit, and leu), alkP, and butyr activities (Figure 5b). In contrast, only the digestate application significantly impacted the respiration of the Tavazzano soil, with a two-fold increase with D++ treatment (Figure 6b). No correlation between soil respiration, dsDNA, and enzymatic activity was observed (Figure 5d).

### 2.4. Impact of Organic Amendments on Leaf Phenolic Compounds

The percentage of N in nettle leaves was not significantly impacted using compost and digestate or by soil (i.e., ~2% on average) (Appendix A). However, the soil (R^2^ = 0.33, *p* = 0.001), the treatment (nature of amendment × dose × soil; R^2^ = 0.28, *p* = 0.001), and, to a lesser extent, the dose of applied amendment (R^2^ = 0.14, *p* < 0.01) significantly impacted the leaf phenolic compound concentration.

Hydroxycinnamic acids (HCAs) and flavonoids constituted the main soluble phenolics detected in the nettle leaves. Caffeoyl esters (i.e., HCA1) and more specifically chlorogenic acid (i.e., CGA; 33.8 ± 2.2% and 27.4 ± 2.0% of total phenolic compounds for the St-Symphorien-sur-Saône and Tavazzano soil, respectively) and caffeoyl malic (i.e., CMA; ~37% of total phenolic compounds for both soils) mainly represented the phenolic compounds (Figure 7a,b). Flavonoids were the second most abundant phenolic compounds (12.90 ± 0.81% and 16.43 ± 1.12% on average for the St-Symphorien-sur-Saône and Tavazzano soils, respectively) followed by p-coumaric esters (i.e., HCA2; 5.8 ± 0.6% and 7.4 ± 1.5% on average for the St-Symphorien-sur-Saône and Tavazzano soils, respectively). Overall, amendment application (except with C+++) significantly halved and divided by four the total phenolic compounds and HCA1 concentration in the leaves of nettles from the St-Symphorien-sur-Saône soil (Figure 7a). C++ and D++ significantly decreased the HCA2 concentration, while only C++ significantly reduced the flavonoid concentration by three times. In contrast, applying amendments to the Tavazzano soil significantly increased the leaf total phenolic compounds and HCA1 concentrations by three- to four-fold (Figure 7b). Only digestate significantly increased and almost tripled the HCA2 concentration (Figure 7b), while all amendment treatments at least significantly doubled the flavonoid concentration.

These observed changes in concentrations did not strongly impact the quantitative abundance of the CMA and flavonoid compounds. In the St-Symphorien-sur-Saône soil, C++ significantly reduced the abundance of CGA by ten and concomitantly increased the abundance of HCA2 by six with D+ treatment (Figure 7a). In the Tavazzano soil, D+ significantly increased the abundance of CGA by more than 10% but decreased the percentage of HCA2 by three to four along with compost (Figure 7b).

## 3. Discussion

### 3.1. Urtica dioica Can Be Grown in Hg-Contaminated Soils, but Its Growth Is Impacted by Soil Properties

*Urtica dioica* grows spontaneously in contaminated lands but its cultivation in such soils has not been previously studied. However, the contamination by inducing stress can affect plant growth and physiology [64,65]. In this context, assessment and improvement of plant growth and physiological traits are important [66]. In our study, stinging nettle grew well in Hg-contaminated soils, with a photosystem II photochemical efficiency slightly below 0.8, suggesting that the nettles were not stressed by soil conditions [67,68]. The two soils investigated in the present study exhibited different soil properties, particularly in terms of Hg concentration, which was six times higher for the Tavazzano soil. However, the levels of Hg found in the tissues of nettles grown in both soils were similar to those measured in crops (e.g., 6–139 μg·kg^-1^, [31]), suggesting that Hg was poorly transferred to the aboveground part of the plant, as previously described [29,69,70]. However, the Hg levels were approximately five times greater than those found in our previous study [71], which is probably because the nettle leaves from the present study were much younger, closer to the soil, and probably more active in terms of Hg uptake.

The stinging nettles grown in the Tavazzano soil produced overall more biomass, with higher plants and greater stem diameters than the nettles from the St-Symphorien-sur-Saône soil. Soil is the main driver of nettle growth, and although the Tavazzano soil is slightly lower in N, it contains more phosphorus, more clay, and two times less organic matter than the St-Symphorien-sur-Saône soil. This suggests that, probably more than N, described as the most important element in stinging nettle nutrition [72], P and soil texture are important parameters for the growth and development of *U. dioica*.

### 3.2. Solid Digestate Has a Greater Impact Than Compost on Stinging Nettle Growth

Compost and solid digestate have been described as effective amendments to improve plant growth [73,74]. Nevertheless, the effects of digestate described in the literature are highly variable due to the dose, fraction (i.e., liquid or solid), the type of feedstock used for digestion, and the type of soil studied [75]. In our study, the application of digestate and compost had different impacts. The two higher doses of amendments (C+++ and D++) increased the biomass, relative growth, plant height, stem diameter and length, and number of internodes of nettle grown in the St-Symphorien-sur-Saône soil. In the Tavazzano soil, only digestate significantly improved plant growth parameters. Overall, the nature of the amendments had a great impact on plant parameters since the effect of compost differed according to the soil type [76]. Notably, D++ significantly increased all measured parameters compared to compost while providing half as much N. In the St-Symphorien-sur-Saône soil, D+ significantly doubled plant biomass, as did C+++, while C++ did not improve biomass. Thus, much more compost than digestate would be needed to improve plant parameters at this site. These results confirm the lower impact of compost on plant growth and biomass in contaminated soils [77,78] when compared to digestate [79]. This may be due to the composition of the digestate, which contains more directly available plant N than the compost [80,81]. In addition, although the compost used contained more organic matter than the solid digestate, this amendment was produced from plant material. Therefore, this OM is partly composed of lignin, which is hardly biodegradable [81]. N is therefore mainly present in organic form and thus more slowly released and available to plants [82]. Mineralization of compost may be a longer process whereas stinging nettle has a high growth rate. Nitrogen release is therefore not synchronous with plant N uptake, as previously demonstrated for sweet corn [83]. Another possibility for the low impact of compost on plant growth may be the maturity of the amendment [40]. N immobilization can indeed occur quickly in less mature composts [84]. As a consequence, microorganisms scavenge available N from the soil that becomes unavailable to plants [85,86], explaining that nettles were similar to the control in terms of plant parameters, without N accumulation in aerial parts but with higher microbial biomass in all treatments in the St-Symphorien-sur-Saône soil.

Compost application had no impact on nettles grown in the Tavazzano soil, but it can provide a longer-lasting effect [87]. Compost thus does not appear to be suitable for improving the growth of *U. dioica* in soils with low nutrient availability in the short term. The low impact of compost on plant yield following the first application has already been described in the case of fertile soils [55,83], but further investigations are needed to assess its potential effect in the long term. In addition, relative growth was negatively impacted by digestate during the first days of cultivation. Temporary immobilization of soil N has already been described during the use of organic amendments [88] and may be responsible for this slowing in plant growth. It can therefore be recommended to add the amendments two weeks before planting the nettles. Nevertheless, the nettles have started to grow despite the low availability of nutrients, probably thanks to their rhizomes, which constitute an N reserve that can be mobilized primarily for shoot development [89]. Finally, the relative growth of *U. dioica* was enhanced using amendments after several weeks of cultivation (i.e., 63 days).

### 3.3. The Dose of Amendment Applied Impacts the Growth of the Stinging Nettle and Reduces the Concentration of Hg in the Aerial Parts

Although described as a crop with low requirements for agricultural soils, the use of quite high fertilization doses has been reported for *U. dioica* cultivation [32,34]. In the present study, nettle biomass followed the dose of amendment applied. However, high doses of amendments were used in this study to double the nettle biomass. Growing *U. dioica* on marginal lands seems to require high inputs to enhance its biomass, which should be considered regarding environmental concerns (e.g., gaseous emission, nutrients leaching, eutrophication) [90,91]. Moreover, under the highest fertilization dose of digestate, the nettle plants all reached the flowering stage, while flowers only started to appear in the other treatments except for the control. As the flowering stage induces the cessation of vegetative growth [92], the digestate significantly improved the biomass produced but reduced the growth period.

In addition to impacting the biomass, the production of fibers was also enhanced without impact on the fiber size, which was similar to those previously reported and therefore similar to fibers of hemp (*Cannabis sativa* L.) and flax (*Linum usitatissimum* L.) [37]. As nettle fibers are potentially relevant candidates for composite reinforcement applications [29,93], improving nettle traits without TE accumulation through fertilization can be valuable for biomass production from contaminated lands. Organic amendments can indeed modulate the mobility and bioavailability of TE [94,95]. In our study, D++ significantly reduced the leaf Hg concentration for both soils, and the stem Hg concentration for the St-Symphorien-sur-Saône soil. Digestate efficiently reduced the mobility of Hg in an artificially contaminated chernozem soil [96], suggesting that solid digestate in high doses can reduce Hg transfer from the soil to the plant by reducing its mobility in the soil. Digestate therefore appears to be a relevant candidate to improve the performance of *U. dioica* grown in contaminated soil, but the dose should be adapted for in situ application.

### 3.4. Organic Amendments Differentially Impact Soil Enzyme Activities Depending on the Soil Type

In addition to its impact on plant parameters, fertilization is also known to induce changes in soil microbial communities and indirectly impact associated enzymatic activities [97,98]. In our study, the application of digestate significantly improved the respiration of the two soils and the microbial biomass of the St-Symphorien-sur-Saône soil, as observed in previous experiments [99,100,101,102]. In contrast, compost application in the Tavazzano soil and C++ application in the St-Symphorien-sur-Saône soil did not significantly impact the soil respiration or the root biomass, which was not significantly different or lower compared to the control plants. This result follows previous studies that highlighted a significant correlation between root biomass and soil respiration [103,104]. Nevertheless, soil respiration was also correlated with dsDNA and enzymatic activities in the St-Symphorien-sur-Saône soil, and amendment application significantly increased the esterase activities in both soils. Butyrate esterase (butyr) has been described to be related to living biomass content [105] and was strongly correlated with dsDNA in the St-Symphorien-sur-Saône soil. Therefore, the enzymatic activities measured in our study were likely related to microorganisms rather than to plants [106]. These results follow previous studies that highlighted higher soil microbial biomass and enzymatic activities in response to the application of amendments to contaminated soils [107,108,109,110]. However, nettle growth was not affected by the increased microbial biomass and higher enzymatic activities under the C++ treatment. This indicates that compost did not improve plant nutrient uptake [111], or that the released nutrients were immobilized or used by soil microorganisms [112,113]. Heijboer et al. [114] highlighted that soil nitrogen retention or microbial immobilization of nitrogen can occur with organic amendments at the expense of plant growth, and this phenomenon can last for a few months [82]. This may explain the fact that only the high doses of amendments impacted plant growth in our study, and that root biomass followed aerial biomass. We hypothesized that when nutrient release was sufficient for soil microorganisms and plants, the stinging nettle produced more roots to take up nutrients.

However, the amendments differentially impacted the enzymatic activities depending on the studied soil after three months of cultivation. In the St-Symphorien-sur-Saône soil, amendments significantly increased almost all measured activities (except those involved in the C cycle), while they were mainly comparable or slightly reduced in the Tavazzano soil. This may be related to the properties of the soil or to the dose of amendments that may have a negative impact on enzymatic activities [60,115,116]. In our study, the soil pH cannot explain the variations in the enzymatic activities [117], as only compost application slightly increased it in the Tavazzano soil. However, since the mineralization of nutrients from amendments depends on soil properties and may be faster in more fertile soils [118,119], it can be argued that the mineralization rate of amendments was higher in the Tavazzano soil. Thus, all easily degradable organic matter has already been used, resulting in the restoration of soil microbial activity [120]. Previous experiments have shown a return to baseline enzymatic activities 2–3 months after compost application [121]. Additionally, the Tavazzano soil has a higher clay content, which has been described as retaining organic carbon and nitrogen through sorption of organic matter, making them inaccessible to microbes [122,123]. Access to organic matter is therefore easier for microbes in less textured soils such as that of St-Symphorien-sur-Saône, which explains why amendments improved the growth and biomass of *U. dioica* in the St-Symphorien-sur-Saône soil better than in the Tavazzano soil. In addition to improving nettle growth, amendment application also improved soil parameters and conditions that can improve nettle re-establishment and improve re-growth for further harvest.

### 3.5. Fertilization Impacts the Production of Leaf Phenolic Compounds of the Stinging Nettle

In addition to enzymatic activities, organic amendments differentially affected the leaf phenolic compound concentration of *U. dioica* depending on the soil used. The concentrations of total phenolic compounds, HCA2, and flavonoids did not significantly differ between the control plants grown on the two soils. However, while the amendments significantly increased the total phenolic compound concentrations of nettle leaves grown on the Tavazzano soil, they significantly decreased phenolic compounds in the nettle leaves from the St-Symphorien-sur-Saône soil. The dose of amendments did not differentially affect the concentration of total phenolic compounds following the results from Biesiada et al. [124]. In previous studies, the amount of phenolics decreased upon application of amendment due to an improvement in plant growth status [125,126], as described for *U. dioica* following the growth–differentiation balance hypothesis [127]. Thus, biomass accumulation and specialized metabolism are negatively correlated. As a result, lower production of specialized metabolites is observed when plants are cultivated under sufficient nutrients. A greater improvement in plant growth with the amendments was observed in our study on the St-Symphorien-sur-Saône soil compared to the Tavazzano soil.

In contrast, when growth is reduced due to limited nutrient availability, the production of phenolic compounds is enhanced [128,129,130,131]. The accumulation of phenolic compounds is commonly observed as a response to abiotic stress [132] and has been previously described with digestate application [133]. Our results do not follow this rule, as the nettles grown on the Tavazzano soil were not stressed and the application of amendments did not reduce their growth. However, the applied organic amendments contained large amounts of C while N was still limited for the plants. As a consequence, the nettle may have allocated the extra C to the synthesis of phenolic compounds failing to improve its growth [134].

The observed differences in phenolic responses to N fertilization are compound-specific, depending on their biosynthetic pathways [135]. Here, in both soils, hydroxycinnamic acids were the most abundant compounds, as previously described for *U. dioica* [136]. Phenolic acids have been associated with various functions, such as nutrient uptake [137]. The HCA1 concentration was significantly impacted and explained the observed changes in the total phenolic concentration. Flavonoid concentration was impacted, although to a lesser extent, in the nettles from the Tavazzano soil. N limitation has been described as influencing flavonoid biosynthesis [138,139]. When plant nutrient uptake was enhanced, *U. dioica* decreased its phenolic compound production, mainly HCA1, which are the main compounds. In contrast, production is increased when improving growth is not possible, but further investigations on the role of these compounds are still needed.

## 4. Materials and Methods

### 4.1. Collection and Characteristics of the Soils

Two Hg-contaminated soils from St-Symphorien-sur-Saône (47°05′03.9″ N, 5°19′48.7″ E, France) and Tavazzano (45°19′17.0″ N, 9°24′04.1″ E, Italy) were used for the experiment that was conducted in pots. The St-Symphorien-sur-Saône site is a chlor-alkali sediment landfill already described in previous studies [140,141]. The Tavazzano site is a grassland adjacent to a chlor-alkali plant using Hg cell technology. Both soils were sampled at a depth of 0–30 cm, air-dried, and sieved to 4 mm. The total N and C contents were analyzed after dry combustion (Vario Macro Cube, Elementar, Langenselbold, Germany). The soil organic matter content (SOM) was determined by heating the soil sample up to 600 °C (soil TOC cube, Elementar, Langenselbold, Germany). The soil total concentrations of major elements and TE were determined by inductively coupled plasma atomic emission spectrometry (ICP–AES, Thermo Fischer Scientific, Inc., Pittsburgh, PA, USA) after acid digestion of 500 mg of sample (Table 2). The pH of the soil and organic amendments was determined after slow shaking for 30 min at room temperature in a 1:25 soil/1 M KCl solution.

### 4.2. Greenhouse Experimental Design

The company Agrivalor© (Hirsingue, France) supplied the organic amendments prepared from green wastes (compost: C), or from organic food wastes (i.e., food production byproducts, unsold and expired products; solid digestate: D) (Appendix A). These two amendments were mixed with 780 g of sieved soil and placed in one-liter pots. Compost was added to the soil at doses of 0.04% N (i.e., 0.36 g N/kg soil, C++) and 0.07% N (i.e., 0.72 g N/kg soil, C+++), and the solid digestate was added to the soil at doses of 0.02% N (i.e., 0.18 g N·kg^-1^ soil, D+) and 0.04% N (i.e., 0.36 g N·kg^-1^ soil, D++). These doses gave the highest stinging nettle (*Urtica dioica*) biomass improvement based on preliminary experiments (Appendix A). One *U. dioica* plantlet harboring a single rhizome was planted per pot. The High School of Horticulture of Roville-aux-Chênes (Roville-aux-Chênes, France) supplied the nettles, which are produced for multiple uses, including textile fiber. Ten replicates were performed per treatment (i.e., control, C++, C+++, D+, and D++) for each soil. Nettles were grown in a greenhouse for 83 days (e.g., until the first nettles reached the flowering stage) from April to June.

### 4.3. Measured Plant Parameters

Plant height was measured twice a week from Day 0 to Day 83. The leaf chlorophyll content was measured on the second leaf with a chlorophyll content meter (Opti-Sciences, Inc., CCM-200, Tyngsboro, MA, USA) [142]. Photosystem II (PSII) performance was measured on the basis of the fluorescence ratio (F_m_ − F_0_/F_m_) using a MINI-PAM II (Walz, Germany) after placing the plants in the dark for 2 h [143]. At harvest, aerial parts were separated from roots, washed with tap water, and rinsed in distilled water. The stem diameter was measured with a caliper from the middle part of the stem, and the length of internodes was measured.

One internode from the middle of the stem of nettles amended with the highest doses of compost and digestate on St-Symphorien-sur-Saône soil (n = 3 per treatment) was kept in 70% ethanol to characterize the fiber morphology using X-ray microtomography performed on an EasyTom tomograph (RX Solutions, Chavanod, France). The stem pieces were placed vertically in an Eppendorf tube in ethanol. The Eppendorf tube was glued on the sample holder to avoid movement during acquisition. The sample holder was mounted on a rotating stage to allow for a rotation of 360°. The X-ray source, a Hamamatsu Open Type Microfocus L10711 (RX Solution, Chavanod, France), was operated with an electron current of 84 μA and a tube voltage of 60 kV. The X-ray transmission images were acquired using a 2530DX detector of 2176 × 1792 pixels^2^. The exposure time and the average frame were fixed at 1.5 s and 2 images, respectively, and 1440 images per revolution were acquired. The entire volume was reconstructed at a full resolution with a voxel size of 2 μm corresponding to a field of view of 3.8 × 2.8 mm^2^, using filtered back-projection. The data analysis was processed using VG StudioMax software. The analysis of the grey levels was achieved using a threshold to separate the fiber wall from the air and access the geometry of the fiber. These images were then post-processed in MATLAB software. The first step consisted of automatically closing the open contours and removing particles smaller than 5 μm. Then, each fiber was identified and processed individually. The minimum and maximum Feret diameters of each fiber were determined. The average wall thickness was also determined using the coordinates of the center of the ellipse.

The third nettle leaves from the top were scanned and the leaf area was measured using ImageJ software (version 1.53r) [144]. Samples were then air-dried at room temperature and weighed for dry biomass prior to subsequent analysis.

### 4.4. Biochemical Analyses on Plant Leaves

Dry leaves were ground using an MM400 Mixer Mill (Retsch, Eragny sur Oise, France) for 1 min at 30 Hz. Composites of two plants were used for subsequent analysis, resulting in n = 5 replicates per treatment for each soil. Two milligrams of leaf powder were used for CHN analysis (FlashEA 1112, ThermoFisher Scientific, Waltham, MA, USA). Soluble phenolics were extracted from 15–20 mg of dry powder with 0.6 mL methanol/H_2_O (4:1) by 10 min of sonication, followed by shaking for 10 min in the dark. The extraction was repeated twice, and all supernatants were pooled. Trans-cinnamic acid was used as internal standard. Extracts were analyzed by UPLC-DAD-ESI-MS/MS (Nexera2, LCMS-8040, Shimadzu, Kyoto, Japan) using a Luna 5 μm C18(2) 100 Å, 250 × 3 mm column (Phenomenex, Torrance, CA, USA) and a C18 guard column with solvent A (10% methanol and 0.2% formic acid) and solvent B (98% methanol and 0.2% formic acid) and the following gradient: 0–1 min of 8% B; 5 min 20% B; 18 min 55% B; 20 min 100% B; 20–26 min of 100% B. The flow rate was 0.35 mL min^−1^, and the column oven temperature was 40 °C). Quantification was performed using UV detection at 320 nm for hydroxycinnamic acids (chlorogenic acid (CGA) as the standard), 360 nm for flavonols (hyperoside as the standard), and 280 nm for t-cinnamic acid (internal standard). MS detection was used for identification using the following conditions: nebulizing gas (N_2_), 3 L min^−1^; drying gas (N_2_), 15 L min^−1^; desolvation line, 250 °C; heat block temperature, 400 °C; and interface voltage 4.5 kV. Finally, Hg quantification was performed from 25 mg of leaves using an AMA-254 (Altec Co, Chotěboř, Czech Republic) as described in Maillard et al. [140].

### 4.5. Analyses of Soil Enzymatic Activities, Microbial Biomass, and Respiration

Fresh rhizospheric soil was sampled for enzymatic activities and soil microbial biomass. Seventeen enzymatic activities (EA) involved in C, N, P, and S cycling were measured according to Ferrarini et al. [145]: α-glucosidase (alfaG, EC 3.2.1.20), β-glucosidase (betaG, EC 3.2.1.21), α-galactosidase (alfaGAL, EC 3.2.1.22), β-galactosidase (beta GAL, EC 3.2.1.23), β-D-glucuronidase (uroni, EC 3.2.1.31), β-1,4-glucanase (cell, EC 3.2.1.4), β-1,4-xylanase (xilo, EC 3.2.1.14), N-acetyl-β-D-glucosaminidase (chit, EC 3.1.3.2), leucine amino-peptidase (leu, EC 3.4.11.1), acid (acP, EC 3.1.3.2) and alkaline phosphomonoesterase (alkP, EC 3.1.3.1), phosphodiesterase (bisP, EC 3.1.4.1), pyrophosphodiesterase (piroP, EC 3.6.1.9), inositol-P phosphatase (inositP, EC 3.1.3.25), arylsulfatase (aryS, EC 3.1.6.1), butyrate esterase (butyr, EC 3.1.1.1), and nonanoate esterase (nona, EC 3.1). Briefly, enzymatic activities were measured from 400 mg of soil added to 1.4 mL of 3% lysozyme solution and glass plus ceramic beads. After shaking and centrifugation, enzymatic activities were quantified fluorometrically from the supernatant and expressed as nanomoles of 4-MUF·g^−1^ soil per hour. Fluorogenic substrates with 7-amino-4-methyl coumarin (AMC; leu) and 4-methyl-umbelliferyl (MUF; other enzymes) were used as fluorophores.

Soil microbial biomass was measured from the double-stranded DNA (dsDNA) concentration using 400 mg of soil in 0.12 M sodium phosphate solution at pH 7.8 as an extraction buffer [146]. PicoGreen reagent (Life Technologies, Carlsbad, CA, USA) was then used to quantify dsDNA, expressed as nanograms of dsDNA g^−1^ dry soil after being corrected for soil moisture content.

Soil microbial respiration was monitored using an Xstream gas analyzer (Emerson, Langenfeld, Germany). At the end of the experiment, 18 g of fresh soil samples adjusted at 63% of the field capacity were placed in tight flasks and incubated at 23 °C. The produced CO_2_ was measured at the end of the experiment and every 4 days, for a total of 16 days using 5 mL aliquots of the flask atmosphere (sampled using a syringe). An empty sealed flask was used as a negative control. The respiration activity was represented by the produced CO_2_ expressed as carbon mass per gram of DW soil per day (μgC·g^−1^·d^−1^).

### 4.6. Statistical Analyses

Statistical analyses and data visualization were performed using R software (version 2022.07.2+576) [147] and were considered significant at *p* < 0.05. All individuals and variables were plotted on a PCA biplot using the FactoMineR package (version 2.4) [148]. Variables were colored according to their contribution to the principal components. The effect of the nature of the amendment, the dose applied, and the soil on plant parameters, enzymatic activities, and phenolic compounds was assessed using a permutational multivariate analysis of variance (PERMANOVA) based on the Euclidean distance matrix using the vegan package (version 2.5-7) [149]. Three-way analyses of variance (ANOVAs) were performed to test the effects of the nature of the amendment, the dose applied, and the soil on the nettle aerial biomass. All the variables were checked for normality distribution (Shapiro–Wilk test) and homoscedasticity (Levene test). For variables that fit to normal distribution, comparisons between soils and treatments were assessed by Tukey’s HSD multiple comparison post-hoc analyses. The other variables were analyzed using the Kruskal function from the agricolae package (version 1.3-1) [150]. A correlation matrix was performed using the corrplot package to assess the correlations between soil respiration, dsDNA, the percentage of N in the leaves, and the enzymatic activities significantly impacted by the addition of amendments.

## 5. Conclusions

*Urtica dioica* was able to grow in the two Hg-contaminated soils of the present study, without any induced stress. The solid digestate in both doses greatly enhanced the biomass of the nettle in contrast to the compost and seems suitable for improving nettle parameters and fiber production. Moreover, digestate application decreased the leaf Hg concentration at high doses. The digestate, and to a lesser extent the compost, improved soil functions, particularly soil respiration and the enzyme activities involved in C, N, P, and S cycling. The use of organic amendments also impacted the leaf synthesis of hydroxycinnamic acids, but their role still needs to be further studied. The plant response differed depending on the soil characteristics, but the application of an appropriate dose of digestate could be proposed as an efficient strategy for the improvement of nettle cultivation on industrial lands. Field cultivation and *U. dioica* re-establishment need further investigation to assess the long-term efficiency of these organic amendments, and the quantity and quality of fibers for industrial use.

## Figures and Tables

**Figure 1 plants-13-02425-f001:**
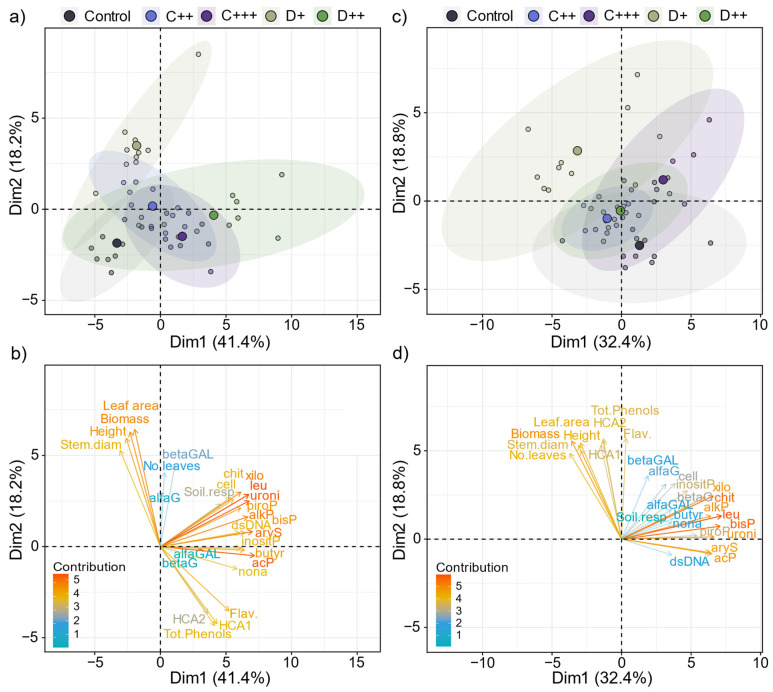
Principal component analysis (PCA) plots showing the ordination of the samples depending on the nature of amendment (C: compost; D: digestate) and rate (+; ++; +++) applied using the whole dataset for (**a**,**b**) St-Symphorien-sur-Saône and (**c**,**d**) Tavazzano soils. The ellipses represent a 95% confidence interval. Vectors are colored depending on their contribution to the overall distribution and indicate the direction and strength of each environmental variable.

**Figure 2 plants-13-02425-f002:**
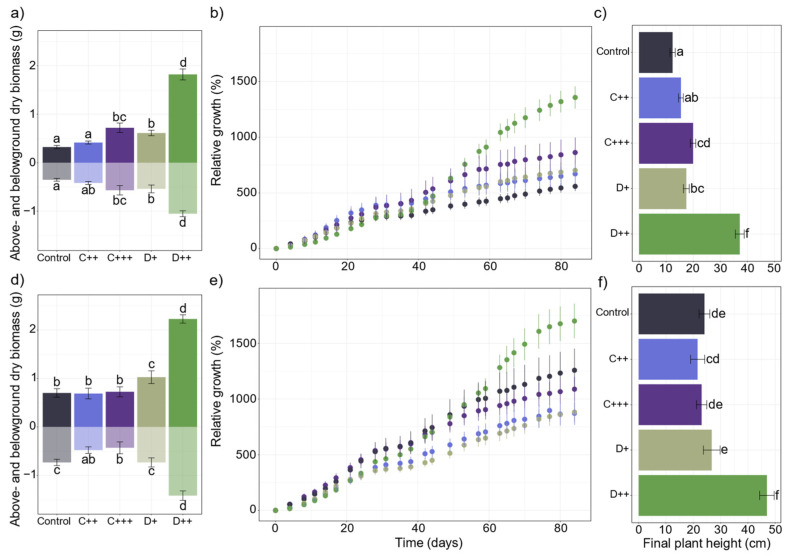
Impact of organic amendments on plant growth parameters (n = 10). (**a**,**d**) Aboveground and belowground dry biomass (g ± SE); (**b**,**e**) relative growth (% ± SE) of nettle plants during cultivation (days) and (**c**,**f**) final plant height (cm ± SE) of *Urtica dioica* after 83 days of cultivation depending on the nature of the amendment (C: compost; D: digestate) and the rate (+; ++; +++) applied to (**a**–**c**) St-Symphorien-sur-Saône and (**d**–**f**) Tavazzano soils. Different letters indicate significant differences between treatments and soils for each variable (Kruskal–Wallis, *p* < 0.05).

**Figure 3 plants-13-02425-f003:**
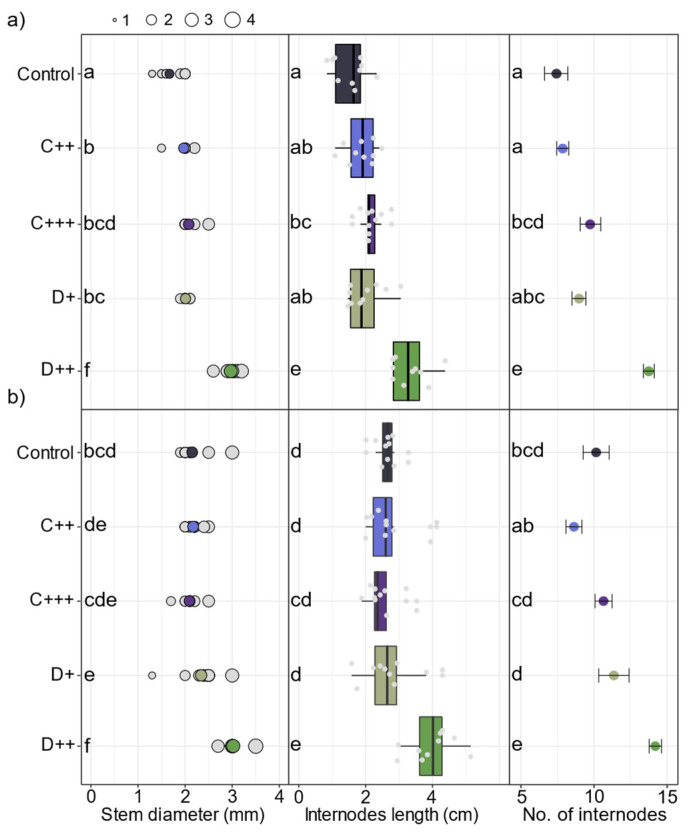
Impact of amendments on nettle stem parameters. Mean stem diameter (mm), internode length (cm), and number of internodes (n = 10 ± SE) per stem after 83 days of cultivation depending on the nature of amendment (C: compost; D: digestate) and rate (+; ++; +++) applied to (**a**) St-Symphorien-sur-Saône and (**b**) Tavazzano soils. Different letters indicate significant differences between treatments and soils for each variable (Kruskal–Wallis test, *p* < 0.05).

**Figure 4 plants-13-02425-f004:**
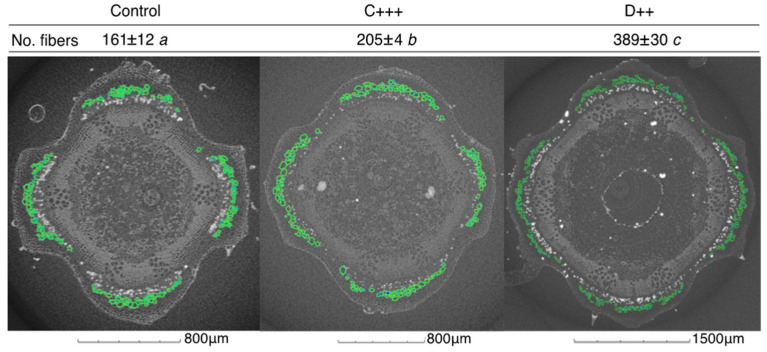
Impact of amendments on the quantity of fibers in nettle stems. Mean number of fibers from *Urtica dioica* stems grown on St-Symphorien-sur-Saône soil (n = 3 ± SE) depending on the amendment applied (C: compost; D: digestate). Pictures represent the associated micro-computed transverse cross-sections of the stems obtained with VG StudioMax software (version 2023.1). Fibers were identified in green with MATLAB software (version R2023b). Different letters indicate significant differences between treatments (Kruskal–Wallis test, *p* < 0.05).

**Figure 5 plants-13-02425-f005:**
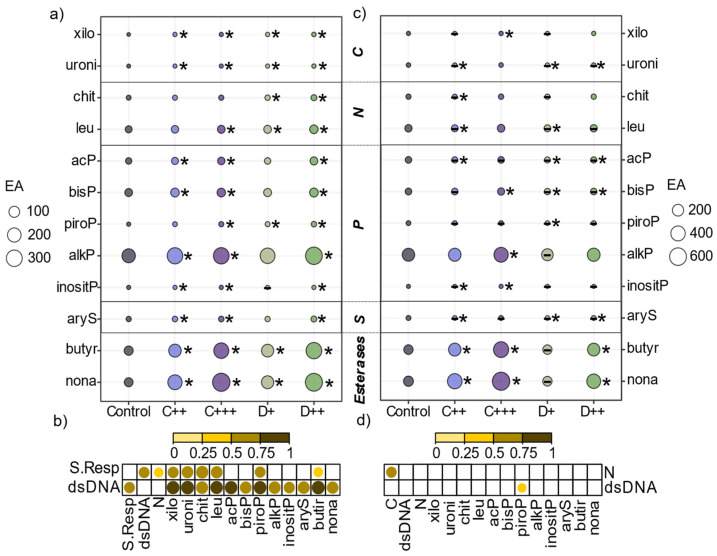
Impact of amendments on rhizospheric soil enzymatic activities. (**a**,**c**) Mean (n = 10) enzymatic activities (EA) expressed in nanomoles of 4-MUF·g^−1^ or AMC·g^−1^ soil per hour involved in carbon (C), nitrogen (N), phosphate (P), and sulfur (S) cycles and esterases depending on the nature of amendment (C: compost; D: Digestate) and rate (+; ++; +++) applied. * indicates a mean significantly different from that of the control treatment for each soil (Kruskal–Wallis or Tukey’s test, *p* < 0.05); - indicates a lower mean than that of the control. (**b**,**d**) Spearman’s correlation matrix between soil respiration (S.Resp), dsDNA, nitrogen (N) concentration, and enzymatic activities in (**a**,**b**) St-Symphorien-sur-Saône and (**c**,**d**) Tavazzano rhizospheric soils. Only the significant correlations are represented.

**Figure 6 plants-13-02425-f006:**
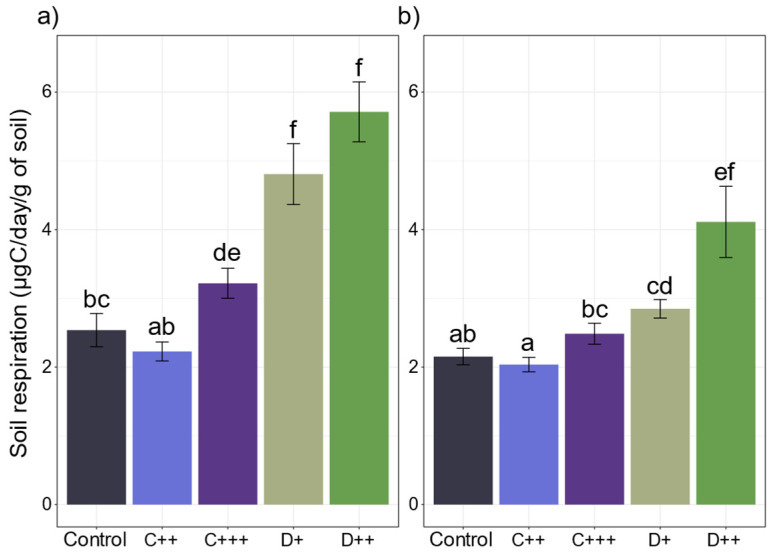
Impact of amendments on soil respiration. Soil respiration (μgC/day/g soil; n = 5 ± SE) after 83 days of cultivation on the (**a**) St-Symphorien-sur-Saône and (**b**) Tavazzano soils depending on the nature of amendment (C: compost; D: digestate) and rate (+; ++; +++) applied. Different letters indicate significant differences between treatments for each soil (Kruskal–Wallis test, *p* < 0.05).

**Figure 7 plants-13-02425-f007:**
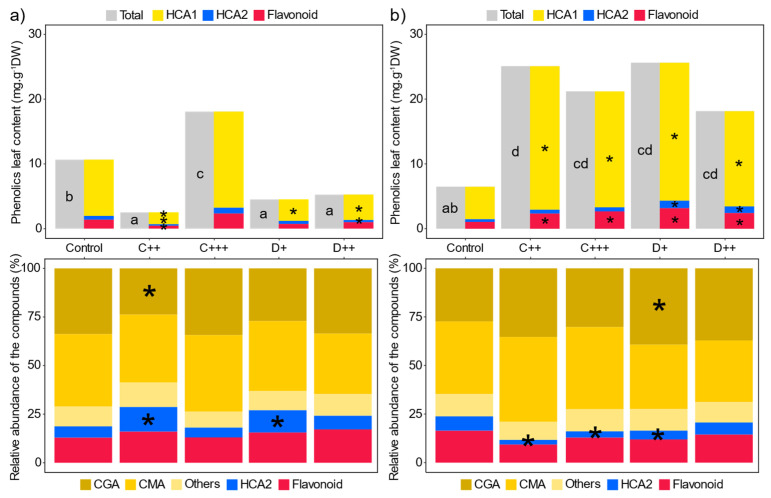
Impact of amendments on leaf phenolic compounds. Mean leaf total phenolic, hydroxycinnamic acid 1 (HCA1), hydroxycinnamic acid 2 (HCA2), and flavonoid concentrations (n = 5, mg·g^−1^DW ± SE) depending on the nature of amendment (C: compost; D: digestate) and rate (+; ++; +++) applied in (**a**) St-Symphorien-sur-Saône and (**b**) Tavazzano soils and their relative quantitative abundances in percentage (chlorogenic acid (CGA), caffeoylmalic acid (CMA) and others are related to HCA1). Different letters indicate significant differences between treatments and soils for total phenolic compounds (Kruskal–Wallis test, *p* < 0.05). * indicates significant differences from the control for each considered compound and for each soil (Tukey’s test, *p* < 0.05).

**Table 1 plants-13-02425-t001:** Mean Hg concentration (n = 5, expressed as μg·kg^−1^ DW ± SE) in the leaves and stems of *Urtica dioica* after 83 days of cultivation on contaminated soils (St-Symphorien-sur-Saône and Tavazzano) depending on the nature of amendment (C: compost; D: digestate) and rate applied (+; ++; +++). Different letters indicate significant differences between treatments and soils (Kruskal–Wallis test, *p* < 0.05).

Soil	Treatment	Leaf Hg Concentration (µg·kg^−1^ DW)	Steam Hg Concentration (µg·kg^−1^ DW)
St-Symphorien-sur-Saône	Control	110.8 ± 15.3 ab	58.8 ± 4.0 efg
	C++	130.2 ± 3.5 a	50.0 ± 6.6 ghi
	C+++	134.5 ± 14.9 a	72.0 ± 10.1 cdef
	D+	101.1 ± 2.7 abc	59.4 ± 8.1 efg
	D++	74.1 ± 14.0 defg	36.0 ± 7.9 hi
	Control	109.3 ± 23.3 abcd	39.9 ± 5.5 hi
	C++	73.5 ± 5.5 bcde	50.2 ± 7.7 ghi
Tavazzano	C+++	152.5 ± 37.2 a	50.9 ± 7.2 fgh
	D+	64.5 ± 11.3 efg	34.5 ± 4.4 hi
	D++	47.6 ± 3.9 ghi	33.2 ± 2.2 i

**Table 2 plants-13-02425-t002:** Properties and elemental composition of St-Symphorien-sur-Saône and Tavazzano soils (mean ± SE).

	Unit	St-Symphorien-sur-Saône (Fr)	Tavazzano (It)
Sand	%	3.8	10.5
Silt	%	87.4	72.5
Clay	%	8.9	16.9
pH		8.0 ± 0.3 × 10^−1^	6.8 ± 0.5 × 10^−1^
C/N		24.0 ± 0.7	18.3 ± 0.6
SOM ^1^	%	4.0 ± 0.1	1.9 ± 0.2
Total N	%	0.1 ± 0.3 × 10^−2^	0.07 ± 0.4 × 10^−2^
TOC ^2^	%	2.3 ± 0.5 × 10^−1^	1.1 ± 0.1
Ca	g/kg	236.8 ± 0.2	3.2 ± 3.7 × 10^−2^
K	g/kg	1.0 ± 1.9 × 10^−2^	2.5 ± 0.2 × 10^−2^
Mg	g/kg	1.3 ± 0.2	4.5 ± 4.0 × 10^−2^
Na	mg/kg	981.8 ± 16.2	84.2 ± 0.8
P	mg/kg	449.0 ± 7.2	753.5 ± 7.3
Fe	g/kg	6.0 ± 7.3 × 10^−2^	14.2 ± 0.1
S	g/kg	13.5 ± 0.1	0.4 ± 0.5 × 10^−2^
As	mg/kg	17.1 ± 0.3	9.2 ± 0.1
Cu	mg/kg	12.6 ± 0.1	16.1 ± 0.6 × 10^−1^
Hg	mg/kg	6.9 ± 0.3	30.0 ± 0.4
Mn	mg/kg	326.1± 6.2	214.1 ± 1.1
Ni	mg/kg	8.7 ± 0.2	26.6 ± 0.2
Pb	mg/kg	17.2 ± 0.8	20.7 ± 0.2
Zn	mg/kg	41.7 ± 0.4	44.5 ± 2.6

^1^ SOM: soil organic matter. ^2^ TOC: total organic carbon.

## Data Availability

Data are contained within the article and Appendix A.

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
