# Peer review of "Digestate Improves Stinging Nettle (Urtica dioica) Growth and Fiber Production at a Chlor-Alkali Site"

_plants, 2024, doi:10.3390/plants13172425_

Round 1

Reviewer 1 Report

Comments and Suggestions for Authors

General comments

The topic investigated is of high interest and actuality, due to the approach adopted, aimed at evaluating the possibility to cultivate U.dioica on TE-contaminated soils to produce fibers. The Authors supported this research with a long list of literature showing their expertise on this plant (genus and species) and the long-lasting activities carried out in the recent years. At the same time from the literature cited, at least one of the chosen contaminated soils is well known and the importance to identify a way to make it productive can be deducted.

The use of organic amendments to enhance the growth of U. dioica is interesting, from one side because it confirmed that digestate reduces the Hg mobility while increasing the biomass production and the number of fibers. On the other side it is evident the impact of soil and the different interactions with respect to nutrients, enzymatic activity and Hg concentrations in leafs and stems.

Specific comments

The two soils used for growing  U. dioica are very different both for texture, physico-chemical and agronomic properties. Both of them are poor in N, thus the additon of compost and digestate is very important for reactivate the microbial activity; probably higher doses of high quality amendments could be used. The soils have an impact on the growth of the plants, which summarizes to the dose and nature of the amendments used. In this sense the high number of analyses carried out strongly support the results and the conclusions even if other aspects remain to clarify due to the complexity of the topic.

I agree with the Authors that digestate seem to be a good candidate but “the dose should be adapted for in situ application” (line 401-402). It could be interesting to transfer in field the adopted approach, to collect more information and test in real scale the effective increase in biomass production of stinging nettle, the quality of fibers for the industrial use, the final destiny of the contaminant (definetly immobilized or other).

Figure 2: very interesting the part a) and c) showing the growth of aboveground and belowground dry biomass, and the effects of digestate on both soils (almost the same). Can compost have a negative effect maybe because not sufficient mature?

Line 329-330: from Table 2 the Tevazzano soil is not slightly richer in N (0,07); please verify.

Did you consider the potential effects of S on the behaviour of the Tevazzano soil?

Did you analyse the soils after the cultivation of plants?

Lines: 350-363: the possibilities you mention to justify the compost behaviour are all real but in some cases they could have been solved by the analysis of the two products used in the study, i.e. compost quality and digestate quality (from yard waste or biowaste), compost maturity (did you measure the respiration rate of compost and digestate?), forms of nitrogen in the two amendments which can cause N immobilization.

Line 351: The “solid digestate” usually is similar to compost for N organic matter content, because the N-NH4 fraction remains in the liquid fraction.  

Line 352: you write that the OM of compost if high because characterized by high lignin content, but the C/N =15 is a good value for a well balanced compost.

Table S5: the N content of compost (9,5) is quite high! Usually compost have about 2% N. It is the double of the digestate (4,2), but you added the double amount of compost to the pots calculated on the %N. Please clarify. Please complete the Table with the unit of measure.

Line 385-387: which environmental concerns do you mean ? Is it elated to the compost or digestate quality ?  Here to remark again the need to have a complete analysis of the two amendment used to exclude possible risks. When obtained form selected organic waste the major concern is the stability and maturity degree, while heavy metals and other contaminants are rare.

Supplemental material

Part of the data and figures refer to the St-Symphorien-sur-Saone, while Tavazzano soil seens to be less known.

Table S5: please see above (specific comments)

Figure S1: the legend is different for a) and b). Please correct.

Figure S2: the part b) of the Figure (Tavazzano data) is lacking. Please complete the figure

Figure S4: the data refer to one type of soil. Did you get the same results for the Tavazzano soil?

Reviewer 2 Report

Comments and Suggestions for Authors Dear authors, i read with interest your manuscript entitled "Digestate improved stinging nettle (Urtica dioica) growth and fiber production at a chlor-alkali site".

The manuscript is well written and results and discussion section are very clear and exhaustive. The introduction needs to be improved as well as the materials and methods part. Some suggestions are given in the attached PDF.

I will gladly read the manuscript again once the changes are made. 

Reviewer 3 Report

Comments and Suggestions for Authors

The paper “Digestate improved stinging nettle (Urtica dioica) growth and fiber production at a chlor-alkali site” studied the growth of Urtica dioica in two mercury contaminated soils with two organic amendments. Plants and soils characterstics were assessed, and the materials and methods session are described in sufficient way with details.

The introduction of the paper presents the current background information but according to my opinion some more details for the uses of U. dioica will be helpful for the readers mainly because the fibers of nettle are also investigated during this study.

The results are presented and explained properly and the references are covered all the sessions of research.

Some minor comments are the below

Line 521: Authors can add some more details for the cultivated nettles, are they a specific variety?

Lines 689-700 are not needed

Reviewer 4 Report

Comments and Suggestions for Authors

The manuscript presents a thorough investigation into the effects of digestate and compost on the growth and fiber production of stinging nettle (Urtica dioica) in Hg-contaminated soils. The experimental design is robust, and the results are well-presented, contributing valuable insights to the field of soil remediation and plant cultivation on industrial lands. However, there are several areas where the manuscript could be improved.

  • Ensure all abbreviations are defined at their first occurrence.
  • There are some typographical errors throughout the manuscript that need correction. For example, in section 2.1, the term "chlorophyll content" is misspelled.
  • References should be checked for formatting consistency, particularly in-text citations and the reference list.

Round 2

Reviewer 2 Report

Comments and Suggestions for Authors

Dear Authors

The manuscript is suitable for publication.

Best Regards